# Analysis of associated malformations by computed tomography in adults with polysplenia syndrome: A pilot study

Xinru Gu[1☯], Shuangshuang Xu[2☯], Jinghua Chen[3☯], Xiaoqin Jiang[1☯], Ping Xie[1☯], Xiang Fang[4], Yan Gao[5], Jian Huang[3], Kefu Liu[1]*

1 Department of Medical Imaging, The Affiliated Suzhou Hospital of Nanjing Medical University, Gusu School of Nanjing Medical University, Suzhou, Jiangsu, China, 2 Department of Radiology, Suzhou Lili Hospital, Suzhou, Jiangsu, China, 3 Department of Radiology, Taicang City Hospital of Traditional Chinese Medicine, Taicang, Jiangsu, China, 4 Department of Radiology, The First Affiliated Hospital of Soochow University, Suzhou, Jiangsu, China, 5 Department of Radiology, The Wuxi Xishan people's hospital, Wuxi, Jiangsu, China

☯ These authors contributed equally to this work.
* lkf77@126.com

**Data Availability Statement:** All relevant data are within the manuscript.

**Funding:** The funding for the research has been provided by " Suzhou key diseases project

## Abstract

### Objective

To analytically depict the associated malformations of polysplenia syndrome (PS) in adults via computed tomography (CT).

### Materials and methods

The incidence of malformations associated with PS in twelve adult patients was retrospectively analyzed via CT imaging.

### Results

The number of splenic nodules ranged from three to twelve; the splenic nodules were located in the left upper quadrant in nine patients and in the right upper quadrant in three patients. A short pancreas was present in all twelve patients. Midgut malrotation was present in eight patients. Situs inversus totalis was present in two patients. Nine patients presented the absence of hepatic segmental inferior vena cava (IVC), with the hepatic vein directly converging into the right atrium and the continuation of the azygos vein. The preduodenal portal vein was present in six patients. Left lung heterotaxy was found in nine patients. The inferior vena cava was bilateral in one patient. Aberrant right subclavian arteries, bilateral common carotid arteries sharing trunks, abnormal renal vein branching and routing, and abdominal portal vein branching were also found in individual patients.

### Conclusions

PS is a complex malformation syndrome involving multiple systems. The most common malformation is short pancreas, and other malformations, such as left lung heterogeneity,

(LCZX202212), Suzhou Science and Technology Bureau Project (SKY2023064, SKYD2023255)" and the funders had no role in study design, data collection and analysis, decision to publish, or preparation of the manuscript.

**Competing interests:** The authors have declared that no competing interests exist.

hepatic segmental IVC agenesis with continuation of the azygos vein, midgut malrotation, preduodenal portal vein, and left atrial heterotaxy, have relatively high prevalence rates.

## Introduction

Polysplenia syndrome (PS), a rare syndrome of congenital anomalous spleen development with other multisystem malformations [1], is also known as left atrial isomerism syndrome (LAIS) because of the frequent occurrence of left atrial isomerism [2]. PS presenting in adults is very rare and has been reported mostly in case studies [3–5]. Compared with other examinations, thoracic and abdominal multilayer computed tomography (CT) is the best means to clarify the details of PS combined with various anatomical variants [6]; therefore, in this study, we retrospectively analyzed the CT imaging data of twelve adults with PS to summarize the concomitant deformities in adults with PS with the aim of improving the understanding of concomitant deformities of PS, avoiding misdiagnosis of anatomical variants as lesions and preventing unnecessary surgical injuries.

## Materials and methods

### Patients

This retrospective study was approved by the institutional review board (No. 2023177, The Affiliated Suzhou Hospital of Nanjing Medical University) with a waiver of written informed consent.

CT images of 12 PS patients from January 2015 to March 2024 were retrospectively collected. The data for this study were accessed from 10/10/2023 to10/10/2024. There were 5 men and 7 women, with an age range of 30–74 years. All 12 patients underwent plain CT examination of the chest and abdomen, and CT enhancement was performed for 6 patients.

### Analysis methods

Two radiologists with 7 and 20 years of experience in diagnostic imaging observed and analyzed the CT images of each patient with PS, respectively, and the concomitant deformities of the PS were recorded.

## Results

### Medical history

A history of cholangiocarcinoma with cirrhosis, cholangiocarcinoma in the porta hepatis with cholecystitis, chronic inflammation of the bile ducts and duodenum, biliary stones, arrhythmia and right kidney stones, left kidney stones, polyps of the colon, and ventricular septal defect (VSD) were recorded in one patient each, and the other patients did not have any special medical history (Table 1).

### CT imaging findings

The concomitant deformities of the PS of 12 patients shown on CT are summarized in Table 2.

Respiratory system: Nine patients (75%, 9/12) presented with left lung heterogeneity (i.e., both the lungs and bilateral bronchi presented with left-sided morphology) (Fig 1a).

Spleen: The number of splenic nodules ranged from three to twelve, with sizes ranging from 3×4×6 mm to 18×30×88 mm (Fig 1b). Polysplenia nodules were located in the left upper

**Table 1. Clinical data of 12 patients with polysplenic syndrome.**

| Case No. | Age (Year) | Sex | Medical history |
|---|---|---|---|
| 1 | 70 | F | Polyp of the colon |
| 2 | 38 | M | VSD |
| 3 | 53 | F | None |
| 4 | 38 | F | None |
| 5 | 74 | M | Cholangiocarcinoma, cirrhosis |
| 6 | 69 | F | Inflammation of the bile ducts and duodenum |
| 7 | 64 | F | Gallstones |
| 8 | 66 | M | Cholangiocarcinoma, Cholecystitis |
| 9 | 31 | M | Arrhythmia, right kidney stone |
| 10 | 30 | M | None |
| 11 | 57 | F | Left kidney stone |
| 12 | 65 | F | None |

VSD- ventricular septal defect

quadrant in nine patients (75%, 9/12) and in the right upper quadrant (25%, 3/12) in three patients (Fig 1b). Splenic nodules were distributed along the external and posterior parts of the greater curvature of the stomach (Fig 1b).

Pancreas: A short pancreas was observed in all twelve patients (100%, 12/12) (the head and uncinate process of the pancreas were enlarged, and the body and tail of the pancreas were short or absent) (Fig 1c).

Hepatobiliary system: Six patients exhibited normal anatomy (50%, 6/12). The liver and gallbladder were found in the left upper quadrant in two patients (17%, 2/12). A small left lobe of the liver was detected in one patient (8%, 1/12), a horizontal liver with a right gallbladder was detected in two patients (17%, 2/12) (Fig 1d), and a horizontal liver and median gallbladder were detected in one patient (8%, 1/12) (Fig 1e).

Gastrointestinal tract: The stomach was located on the right side (Fig 1c) in three patients (25%, 3/12), and midgut malrotation was found in eight patients (67%, 8/12) (Fig 1f).

Heart: Dextrocardia was found in two patients (17%, 2/12) (Fig 1g).

Atrium: CT suggested possible left atrial heterotaxy in five patients (42%, 5/12) (Fig 1h).

Aortic arch: A right aortic arch (RAA) and dextral pulmonary trunk were observed in two patients (17%, 2/12) (Fig 1i).

Superior vena cava (SVC): A persistent left superior vena cava (PLSVC) was found in one patient (8%, 1/12).

Inferior vena cava (IVC): The absence of the hepatic segment of the IVC (Fig 1j and 1k), which continued upward as a thickened azygos vein was found in nine patients (75%, 9/12) (Fig 1k), and a bilateral IVC was found in one patient (8%, 1/12) (Fig 1f).

Portal vein: The preduodenal portal vein (PDPV) was present in six patients (50%, 6/12) (Fig 1l). In one patient (8%, 1/12), the portal vein traveled in front of the head of the pancreas and then bifurcated into two branches; the thicker branch traveled along the anterior border of the liver to the diaphragmatic surface of the liver into the liver, and the thinner branch entered the liver in the first hepatic portal (Fig 1m).

Hepatic vein: In nine patients (75%, 9/12), the hepatic vein exited the second porta hepatis and converged directly into the right atrium (Fig 1k).

Renal veins: Two patients (17%, 2/12) had unilateral renal veins with double-branched variants: one patient had both renal veins converging directly into the azygos vein (Fig 1n), one

**Table 2. The malformations of 12 patients with polysplenic syndrome on CT.**

| Malformations | Percentage of Occurrence | Case1 | Case2 | Case3 | Case4 | Case5 | Case6 | Case7 | Case8 | Case9 | Case10 | Case11 | Case 12 |
|---|---|---|---|---|---|---|---|---|---|---|---|---|---|
| Left lung heterogeneity | 75% | Yes | Yes | Yes | Yes | Yes | No | Yes | Yes | No | No | Yes | Yes |
| Left atrial heterotaxy | 42% | Yes | Yes | No | No | Yes | No | Yes | Yes | No | No | No | No |
| Location of spleen |  | Left | Left | Right | Left | Left | Left | Left | Left | Left | Left | Right | Right |
| Number of spleens |  | 5 | 12 | 4 | 7 | 4 | 3 | 11 | 5 | 12 | 5 | 8 | 5 |
| Liver: small left lobe | 8% | Yes | No | No | No | No | No | No | No | No | No | No | No |
| Left liver | 17% | No | No | Yes | No | No | No | No | No | No | No | No | Yes |
| Horizontal liver |  | No | No | No | No | No | No | Yes | Yes | No | No | Yes | No |
| Location of gallbladder |  | Right | Right | Left | Right | Right | Right | Right | Median | Right | Right | Right | Left |
| Short pancreas | 100% | Yes | Yes | Yes | Yes | Yes | Yes | Yes | Yes | Yes | Yes | Yes | Yes |
| Midgut malrotation | 67% | Yes | Yes | Yes | No | Yes | No | Yes | Yes | Yes | No | Yes | No |
| Hepatic segmental IVC agenesis with continuation of AZ | 75% | Yes | No | No | Yes | Yes | No | Yes | Yes | Yes | Yes | Yes | Yes |
| Bilateral IVC | 8% | No | No | Yes | No | No | No | No | No | No | No | No | No |
| PDPV | 42% | No | Yes | No | Yes | Yes | No | No | No | Yes | Yes | Yes | No |
| Double LRV | 8% | Yes | No | No | No | No | No | No | No | No | No | No | No |
| Double RRV | 8% | No | No | Yes | No | No | No | No | No | No | No | No | No |
| Bilateral RV inflow AZ | 8% | No | No | No | No | No | No | No | Yes | No | No | No | No |
| LRV travels behind AA, and bilateral RV flow into AZ | 8% | No | No | No | No | No | No | No | No | Yes | No | No | No |
| RRV flows into AZ, LRV flows into HAZ | 17% | No | No | No | No | No | No | No | No | No | No | Yes | Yes |
| Bilateral CCA shared trunks | 17% | Yes | No | No | No | No | No | No | Yes | No | No | No | No |
| ARSA | 17% | Yes | No | No | No | No | No | No | Yes | No | No | No | No |
| RAA | 17% | No | No | Yes | No | No | No | No | Yes | No | No | No | No |
| DPT | 17% | No | No | Yes | No | No | No | No | Yes | No | No | No | No |
| Dextrocardia | 17% | No | No | Yes | No | No | No | No | Yes | No | No | No | No |
| SIT | 17% | No | No | Yes | No | No | No | No | No | No | No | No | Yes |
| The anterior branch of the portal vein enters the liver from the anterior edge of the liver | 8% | Yes | No | No | No | No | No | No | No | No | No | No | No |

IVC-inferior vena cava; AZ-azygos vein; PDPV-preduodenal portal vein; RV-renal vein; LRV-left renal vein; RRV- right renal vein; AA-abdominal aorta; SVC-superior vena cava; HAZ-hemiazygos vein; ARSA- aberrant right subclavian artery; CCA-common carotid artery; SIT-Situs inversus totalis; RAA-right aortic arch; DPT-dextral pulmonary trunk

patient had the left renal vein traveling posterior to the abdominal aorta (Fig 1o), and one patient had the right renal vein flowing back into the azygos vein and injecting upward into the SVC. In two patients (17%, 2/12), the left renal vein first converged into the hemiazygos vein and then into the azygos vein in the posterior mediastinum.

Common carotid artery (CCA): Bilateral CCAs shared trunks in two patients (17%, 2/12) (Fig 1p).

Others: An aberrant right subclavian artery (ARSA) was found in two patients (17%, 2/12) (Fig 1p).

## Discussion

### Pathogenesis and survival rates

PS was first reported by Baillie in 1788 [7], but the exact cause of PS is still unclear, and studies have shown that its occurrence is closely related to embryonic development, inheritance,

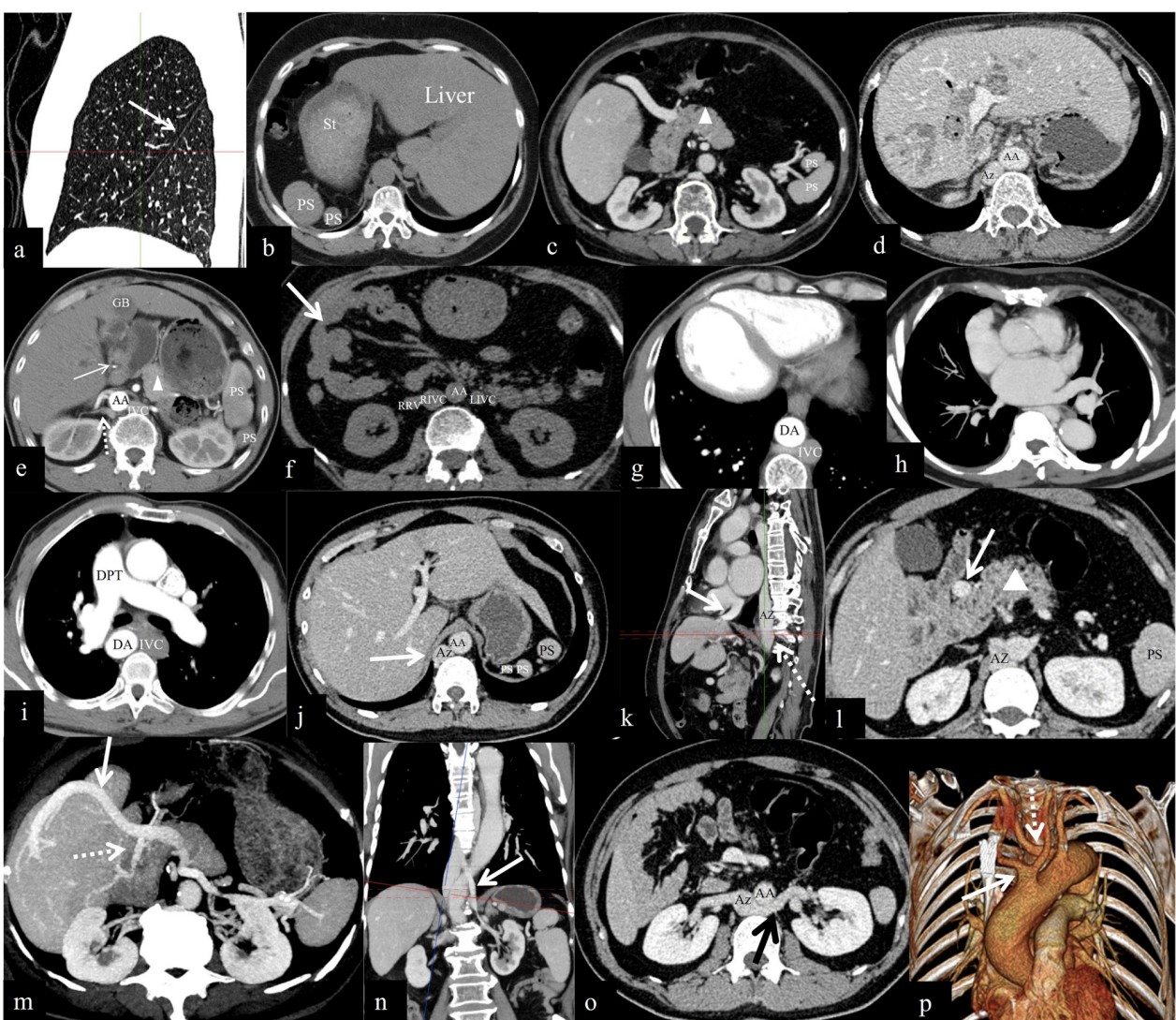

**Fig 1.** (a) Left lung heterogeneity: Sagittal image of the right lung showing only the major oblique fissure (arrow). (b) Visceral inversion: the liver is located on the left side, the stomach (St) is on the right side, and multiple splenic nodules (PSs) are present in the right upper quadrant. (c) A short pancreas (Δ). (d) Horizontal liver. (e) Hepatic hilar cholangiocarcinoma (arrow) and median gallbladder (GB). (f) Malrotation of the midgut (arrow). (g) Dextrocardia. (h) Left atrial heterotaxy. (i) Dextral pulmonary trunk (DPT); the IVC is on the left, and the descending aorta (DA) is on the right. (j) The thickening AZ (arrow) and absence of the IVC of the hepatic segment. (k) The hepatic segment of the IVC is absent, and the left, middle, and right hepatic veins converge and are injected directly into the right atrium (arrow); a thickened AZ (dotted arrow). (l) Preduodenal portal vein (PDPV) (arrow). (m) The portal vein travels anteriorly to the head of the pancreas and then bifurcates into two branches, with the thicker branch traveling along the anterior edge of the liver to the diaphragmatic surface of the liver to enter the liver (solid arrow) and the thinner branch entering the liver in the first hepatic hilar (dotted arrow). (n) Left renal veins converging directly into the azygos vein (arrow). (o) The left renal vein crosses the abdominal aorta (AA) posteriorly to join the Az (arrow), and the right renal vein joins directly to the Az; malrotation of the midgut. (p) Bilateral common carotid artery shared trunks (arrow). Aberrant right subclavian artery (dotted arrow).

teratogenic factors, genetic mutations and other factors [6]. The 5th–7th week of embryo development is a critical stage for the development and rotation of the spleen, atrial septum, trunk of the conus artery and atrioventricular valve, as well as the process of the gastrointestinal tract returning to the abdominal cavity from the umbilical cord for rotation. Cardiovascular and visceral anomalies occur when this process is impaired. Situs solitus describes the normal anatomic position of organs or structures. Situs inversus refers to a left–right

inversion, resulting in the mirror image of situs solitus. Situs ambiguous, a synonym for left or right isomerism, characterizes the positions of organs or structures lateralizing across the left–right axis and cannot be categorized as situs solitus or inversus. Any type of situs other than situs solitus can be described as heterotaxy [8].

PS is very rare, with an infantile morbidity rate of only 1/250000, and the degree of congenital heart disease malformation affects the survival of patients with PS. The mortality rate in the first year is 50–60%; 25% of patients with PS survive to the age of 5 years, and only 10% have normal hearts or only minor heart defects and therefore survive to adulthood [9–11]. Some studies have reported [4, 9] that the prevalence of PS is greater in females than in males, and our findings are consistent with the literature.

## CT findings

**Multiple spleens.**   The number of spleens reported in the literature is between two and sixteen [10], with different sizes, no main spleen, no increase in the total weight of the spleen, and no hypersplenism, and the spleen can be located in the left upper quadrant, right upper quadrant, or both sides of the abdomen, with the left upper quadrant being the most common. The spleen is usually located on the side of the greater curvature of the stomach and shows "Zebra pattern" enhancement in the arterial phase and uniform enhancement in the venous and delayed phases. The degree of enhancement is consistent with that of a normal spleen [6, 11].

The number of spleens in this group ranged from three to twelve, twelve of which were located next to the greater curvature of the stomach and, three of which were located in the right upper quadrant. Six patients who completed the CT enhancement examination had an arterial "Zebra pattern" and uniform venous enhancement, twelve patients had no main spleen, and the size of the splenic nodules varied, which was consistent with previous reports.

**Visceral abnormalities.**   In our group, all twelve patients had short pancreas, including two patients with pancreatic inversion, which was the most prevalent extrasplenic abnormality. Some studies [12–15] have reported that pancreatic abnormalities are associated with PS. The pancreas is formed by the fusion of ventral and dorsal pancreatic buds; the ventral pancreatic buds give rise to the uncinate process and head, and the dorsal pancreatic buds give rise to the body and tail [14]. When the pancreas is dorsally underdeveloped, dysplasia of the body and tail of the pancreas, occurs which mostly manifests as a short pancreas [14].

In this group, one patient had a small left lobe of the liver accompanied by a right gallbladder, two patients had total visceral inversion manifesting as a left liver and gallbladder, two patients had a horizontal liver with a right gallbladder, and one patient had a horizontal liver with a median gallbladder. Some previous studies have also reported these hepatobiliary anomalies [4, 12].

In our group, there were eight cases of midgut malrotation (8/12, 67%), including two cases of gastrointestinal anteversion (total visceral anteversion), which is consistent with the published literature showing small bowel malrotation in 60.4% of PS patients [4, 16].

In our group, we found kidney stones in two of our twelve patients, and no other genitourinary anomalies, such as a double ureter or renal hypoplasia [17, 18], were observed.

**Vascular malformations.**   In our group, hepatic segmental IVC agenesis with continuation of azygos was found in nine patients. When the hepatic segment of the IVC fails to anastomose with the suprarenal segment during embryonic development, the renal segment and lower segment flow back into the SVC via the azygos or hemiazygos vein [6], which is considered the most common vascular malformation in PS [9, 19, 20].

In our group, PDPV was present in six patients. PDPV [21, 22] is a rare congenital anomaly, especially in adults, in which the portal vein passes through the anterior part of the duodenum

rather than the posterior part and leaves an abnormal vein in the anterior part of the duodenum due to an anomaly of rotation during intestinal rotation or an anomaly of closure of the vitelline vein (umbilical vein), which results in the formation of a portal vein. In patients who require surgery, failure to detect the presence of PDPV before surgery may lead to serious complications such as intraoperative bleeding. In addition, it has been reported [23] that PDPV may compress the duodenum and biliary tract, causing intestinal obstruction and biliary obstruction.

Renal vein anomalies mainly manifest as the retro-aortic renal vein and the right renal vein flowing back into the azygos vein to inject upward into the SVC, and the left renal vein converges into the hemiazygos vein, which converges into the azygos vein in the posterior mediastinum [4]. Among them, the retro-aortic renal vein usually occurs unilaterally on the left side, which can lead to hematuria, left-sided pain, and a variety of other symptoms, often suggesting posterior nutcracker syndrome [24], which is relatively rare and is consistent with the presentation in one patient in our group.

Other vascular malformations, including ARSA and bilateral CCA shared trunks [4], were found in our group, with the exception of vertebral artery origin anomalies. In our study, two special manifestations were observed that were not reported in the previous PS-related literatures: bilateral IVC and portal vein abnormalities bifurcated into two branches with the thicker branch traveling along the anterior border of the liver to the diaphragmatic surface of the liver to enter the liver and the thinner branch entering the liver at the first hepatic hilar.

**Lung anomalies.**   Nine patients in our group exhibited left lung heterogeneity, with a higher incidence than that reported in previous studies, which reported an incidence of 55% [10]. A previous study [6] also revealed that the incidence of bilateral lung heterogeneity on the left side was much lower in adults with PS, which suggested that since adults with PS usually do not have cardiac anomalies such as abnormal pulmonary venous return or atrial septal defect (ASD), they do not experience bilateral left-sided changes. In our group, there were only two patients with cardiac abnormalities, which is not consistent with previous reports in the literature [6, 10]. The reason for this may be that left lung heterogeneity is related not only to cardiac abnormalities but also to other factors that need further confirmation.

**Cardiac abnormalities.**   In this group, five CT scans suggested the possibility of left atrial heterotaxy, one had a history of VSD, and cardiac ultrasound suggested a tumor in the membranous portion of the interventricular septum. PS in adults is usually not associated with cardiac developmental abnormalities or only with mild cardiac abnormalities [3]. These abnormalities mainly include positional abnormalities of the heart, such as dextrocardia [9]; structural abnormalities, such as ASD, VSD, right ventricular double outlet, atrial septal hypertrophy, and lipoma [11]; and arrhythmias, such as atrioventricular block and bradycardia [25], some of which are difficult to diagnose via CT and usually rely on cardiac ultrasound and electrocardiogram.

**Comorbid tumors.**   In our group, there were two patients with cholangiocellular carcinoma, which was also reported previously [26], and the cause is hypothesized to be related to biliary obstruction due to developmental malformations of the biliopancreatic duct and recurrent infections of the biliary system in patients with PS. Previous case reports [15, 26–28] have shown that PS can co-occur with malignant tumors, such as hepatocellular carcinoma, intrahepatic cholangiocarcinoma, gastric carcinoma, rectal carcinoma, and breast cancer. However, there is no evidence that PS is a precancerous syndrome or that there is a greater incidence of tumors than in the normal population.

**Differential diagnosis.**   PS needs to be differentiated from asplenia syndrome (AS), accessory spleen, simple visceral inversion, enlarged lymph nodes in the abdominal cavity, and other nodular masses in the abdominal cavity [7, 9, 29]. (1) AS can present as congenital

splenic hypoplasia or absence of the spleen, with trilobarism of both lungs (right-sided heterotaxy of both lungs), a median liver, and more frequent and more severe cardiovascular malformation than PS, and the IVC is located on the same side of the spine as the aorta. (2) Accessory spleen: There is a large primary spleen, with a large difference in volume between the primary and secondary spleens, whereas there is no primary spleen in PS. (3) Simple inversion of the viscera: partial or total inversion of the viscera, resembling the mirror image of a normal person, is not associated with cardiovascular or visceral developmental malformations. (4) Enlarged lymph nodes in the abdominal cavity and other nodular masses in the abdominal cavity: There is a history of inflammation or malignant tumors, often multiple, located in the abdominal cavity and retroperitoneum, without a splenic hilar incision or splenic hilar vessels, and homogeneous enhancement in all three phases is observed.

## Conclusions

PS is a complex malformation syndrome involving multiple systems. The most common malformation is a short pancreas, and other malformations, such as left lung heterogeneity, hepatic segmental IVC agenesis with continuation of the azygos vein, midgut malrotation, preduodenal portal vein, and left atrial heterotaxy, have relatively high prevalence.

Being familiar with the occurrence of malformations associated with PS is conducive to avoiding misdiagnosis and mistreatment. Because the malformations involve multiple systems, multidisciplinary consultation is more conducive to the evaluation of this disease.

## Author Contributions

**Conceptualization:** Kefu Liu.

**Data curation:** Shuangshuang Xu, Jinghua Chen, Xiaoqin Jiang, Ping Xie, Xiang Fang, Yan Gao, Jian Huang, Kefu Liu.

**Formal analysis:** Xinru Gu.

**Funding acquisition:** Kefu Liu.

**Resources:** Xinru Gu, Kefu Liu.

**Writing – original draft:** Xinru Gu, Shuangshuang Xu, Jinghua Chen, Xiaoqin Jiang, Ping Xie, Kefu Liu.

**Writing – review & editing:** Xinru Gu, Shuangshuang Xu, Jinghua Chen, Xiaoqin Jiang, Ping Xie, Kefu Liu.

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
