## [Decision Letter · Decision Letter 0]

29 Aug 2024

PONE-D-24-28190Analysis of associated malformations in adults with polysplenia syndrome on CTPLOS ONE

Dear Dr. liu,

Thank you for submitting your manuscript to PLOS ONE. After careful consideration, we feel that it has merit but does not fully meet PLOS ONE’s publication criteria as it currently stands. Therefore, we invite you to submit a revised version of the manuscript that addresses the points raised during the review process.

**Please review the comments by reviewers and make the appropriate revisions.**

We look forward to receiving your revised manuscript.

Kind regards,

Gurinder Kumar, MD

Academic Editor

PLOS ONE

**Journal Requirements:**

6. PLOS requires an ORCID iD for the corresponding author in Editorial Manager on papers submitted after December 6th, 2016. Please ensure that you have an ORCID iD and that it is validated in Editorial Manager. To do this, go to ‘Update my Information’ (in the upper left-hand corner of the main menu), and click on the Fetch/Validate link next to the ORCID field. This will take you to the ORCID site and allow you to create a new iD or authenticate a pre-existing iD in Editorial Manager.

Reviewers' comments:

Reviewer's Responses to Questions

**Comments to the Author**

1. Is the manuscript technically sound, and do the data support the conclusions?

Reviewer #1: Yes

Reviewer #2: Yes

Reviewer #3: Yes

Reviewer #4: Yes

2. Has the statistical analysis been performed appropriately and rigorously? 

Reviewer #1: Yes

Reviewer #2: No

Reviewer #3: Yes

Reviewer #4: N/A

3. Have the authors made all data underlying the findings in their manuscript fully available?

Reviewer #1: Yes

Reviewer #2: Yes

Reviewer #3: Yes

Reviewer #4: No

4. Is the manuscript presented in an intelligible fashion and written in standard English?

Reviewer #1: Yes

Reviewer #2: Yes

Reviewer #3: Yes

Reviewer #4: Yes

5. Review Comments to the Author

**Reviewer #1: **Rigorously written article by the authors. It is a retrospective series, however the rarity of the condition makes the article useful. Adding percentages of findings in tables would have added to the readability of the article.

**Reviewer #2:** Dear Author,

Greetingd to you for drafting a study on an important disorder such as the Polysplenia disorder.

I commend your efforts totally.

I request you to please go through these observations at my end.

1. You can shorten the running title as the main title and running title appear to be the same.

2. Keywords may be chosen a bit more appropriately, for example, you may use a keyword like malformations which may be used instead of 'adult' making it a bit more specific and catchy.

3. The sample size being 12, I request you to make a small change in your title if possible, you can call this as a 'pilot' study.

4. In financial disclosure section, you have disclosed that there was no funding prpvided for your study. Under the fundings head, you are mentioning certain project names. Can you please clarify it further on how the projects helped in your study funding? Was it based on collection of patients or something related?

5. In your objective, can you replace the word 'summarise' to 'analytically depict' so that it becomes more measurable and oriented to your study results.

6. If you can have some statistical tests suitaing this quantitative study wherein you can further supplement your results with statisical evidence, it would make your study stronger.

7. Were the examining radiologists blinded from the patients? Were the patients blinded from the examining radiologists? Were both blinded from each other? Were the examining radiologists aware that this sort of study may take place in future subject to their reporting?

You can go through these points and take your time to consider them as well.

Regards.

**Reviewer #3:** Thank you for providing me the opportunity to review this original article. This article presents an interesting and rare appearance of polysplenia and associated anomalies. A few corrections are required for further processing which have been described in the word file in the comment section.

**Reviewer #4:** With 12 cases, the topic should be better under a case review series rather than a research article.

Author/s should describe the spectrum of heterotaxy or situs ambiguus syndromes in a small paragraph in the discussion introduction.

The author/s should mention in a concise paragraph the need for symptomatic management in the patients, depending on the type of associated anomaly, along with the importance of frequent follow-up with a multidisciplinary team. 

Fig 1l Duodenal duodenum anterior portal vein (PDPV) - correct the terminology

6. PLOS authors have the option to publish the peer review history of their article (what does this mean?). If published, this will include your full peer review and any attached files.

Reviewer #1: No

Reviewer #2: **Yes: **Gaurav Vedprakash Mishra

Reviewer #3: **Yes: **Anshul Sood

Reviewer #4: **Yes: **Dr Swati Goyal

---

## [Author Response · Author response to Decision Letter 0]

26 Sep 2024

For review comments

Reviewer1：

1. Rigorously written article by the authors. It is a retrospective series, however the rarity of the condition makes the article useful. Adding percentages of findings in tables would have added to the readability of the article.

Answer：We had added the percentages of findings in table.

Reviewer2:

1. You can shorten the running title as the main title and running title appear to be the same.

Answer：We think this is an excellent suggestion. We have simplified the running title to “Malformations in Adults with Polysplenia on CT”.

2. Keywords may be chosen a bit more appropriately, for example, you may use a keyword like malformations which may be used instead of ＂adult＂ making it a bit more specific and catchy.

Answer：Thanks for your suggestion. We have replaced the keyword “adult” with “malformations”.

3. The sample size being 12, I request you to make a small change in your title if possible, you can call this as a ＂pilot＂ study.

Answer：We agree with your suggestion. We have added “A Pilot Study” in Title. 

4. In financial disclosure section, you have disclosed that there was no funding prpvided for your study. Under the fundings head, you are mentioning certain project names. Can you please clarify it further on how the projects helped in your study funding? Was it based on collection of patients or something related?

Answer：The funders had no role in study design, data collection and analysis, decision to publish, or preparation of the manuscript. We had added this sentence in Statement.

5. In your objective, can you replace the word ＂summarise＂ to ＂analytically depict＂ so that it becomes more measurable and oriented to your study results.

Answer：As suggested by the reviewer, we have replaced the word “summarise” with “analytically depict”.

6. If you can have some statistical tests suitaing this quantitative study wherein you can further supplement your results with statisical evidence, it would make your study stronger.

Answer：Thank you for the valuable suggestions. We had added the percentages of findings in result and table 2.

7. Were the examining radiologists blinded from the patients? Were the patients blinded from the examining radiologists? Were both blinded from each other? Were the examining radiologists aware that this sort of study may take place in future subject to their reporting?

Answer：The study is a retrospective study. We retrospectively analyzed the CT imaging data of twelve adults of PS to summarize the concomitant deformities in adults of PS with the aim of improving the understanding of concomitant deformities of PS, avoiding misdiagnosis of anatomical variants as lesions and causing unnecessary surgical injuries.

Reviewer3:

1. Kindly add full form of CT in the title.

Answer: Thank you for the valuable suggestions. As suggested by the reviewer, we have added the full form of CT (Computed Tomography) in the title.

2. Kindly add an ＂s＂after ＂patient＂.

Answer: Thank you for your detailed review. We have made the necessary corrections to the term “patient”.

3. Kindly correctly label the figure 1b and figure 1c.

Answer: Thanks for this comment. We have corrected the label of the figure 1b and figure 1c.

4. “observed” would be a better term than “affected”.

Answer: Thanks for your suggestion. We have replaced “affected” with “observed” as suggested.

5. Throughout the manuscript, kindly use full form of the acronym when it is used as a first word of the sentence.

Answer: As suggested by the reviewer, we have changed the abbreviations at the beginning of paragraphs throughout the manuscript to their full forms.

6. Kindly use a widely accepted term of enhancement for describing the pattern of enhancement. Something like Zebra pattern.

Answer: We appreciate your suggestion and have made the necessary changes, replacing “blotchy” enhancement with “Zebra pattern” enhancement.

7. Pancreas would suffice.

Answer: Your suggestion is greatly appreciated. We have made the change “Pancreases” to “Pancreas”.

8. Rewrite the sentence ＂Since the normal pancreas...＂ for clarity.

Answer: We sincerely thank the reviewer for careful reading. As suggested by the reviewer, we have revised the sentence to “The pancreas is formed by the fusion of ventral and dorsal pancreatic buds; the ventral pancreatic buds give rise to the uncinate process and head, and the dorsal pancreatic buds give rise to the body and tail [14]. When the pancreas is dorsally underdeveloped, dysplasia of the body and tail of the pancreas, occurs which mostly manifests as a short pancreas [14].”.

9. Kandly identify the site of gall bladder.

Answer: Thank you for your suggestion. We had checked the site of gall bladder.

10. This statement contradicts the statement in the results section and image 1m. Kindly confirm and rewrite.

Answer: We had made a consistent statement for this finding in our paper.

11. Conclusion should be a separate heading and not a subheading. Kindly rectify.

Answer: As suggested by the reviewer, we have revised the heading of Conclusion.

12. Kindly elaborate what does the white arrow in the part (d) indicates.

Answer: It showed the hirizontal liver and biliary stones. To avoid misunderstanding, we had deleted this arrow.

13. Kindly enlarge the dotted arrow in the figure 1k and figure 1m.

Answer: We had enlarged the dotted arrows in the fig 1k and fig 1m.

14. Kindly add arrows to the image 1p.

Answer: We had added the arrows for image 1p.

15. Kindly link the table 2 in the text.

Answer: We had linked the table 2 in the text.

Reviewer4:

1. With 12 cases, the topic should be better under a case review series rather than a research article.

Answer: Thank you for the helpful suggestion. We had added the A PILOT STUDY in Title.

2. Author/s should describe the spectrum of heterotaxy or situs ambiguus syndromes in a small paragraph in the discussion introduction.

Answer: We had added some sentences about heterotaxy or situs ambiguous in the first paragraph in discussion. 

3. The author/s should mention in a concise paragraph the need for symptomatic management in the patients, depending on the type of associated anomaly, along with the importance of frequent follow-up with a multidisciplinary team. 

Answer: We had added a concise paragraph, “Being familiar with the occurrence of malformations associated with PS is conducive to avoiding misdiagnosis and mistreatment. Because the malformations involve multiple systems, multidisciplinary consultation is more conducive to the evaluation of this disease.”, as last paragraph.

4.Fig 1l Duodenal duodenum anterior portal vein (PDPV) - correct the terminology

Answer: Thank you for your suggestion. We have corrected ＂Duodenal duodenum anterior portal vein (PDPV)＂ to ＂Preduodenal portal vein (PDPV)＂.

---

## [Decision Letter · Decision Letter 1]

9 Oct 2024

Analysis of associated malformations by computed tomography in adults with polysplenia syndrome: A Pilot Study

PONE-D-24-28190R1

Dear Dr. liu,

We’re pleased to inform you that your manuscript has been judged scientifically suitable for publication and will be formally accepted for publication once it meets all outstanding technical requirements.

Kind regards,

Gurinder Kumar, MD

Academic Editor

PLOS ONE

Additional Editor Comments (optional):

Reviewers' comments:

Reviewer's Responses to Questions

**Comments to the Author**

1. If the authors have adequately addressed your comments raised in a previous round of review and you feel that this manuscript is now acceptable for publication, you may indicate that here to bypass the “Comments to the Author” section, enter your conflict of interest statement in the “Confidential to Editor” section, and submit your "Accept" recommendation.

Reviewer #2: All comments have been addressed

Reviewer #3: All comments have been addressed

2. Is the manuscript technically sound, and do the data support the conclusions?

Reviewer #2: Yes

Reviewer #3: Yes

3. Has the statistical analysis been performed appropriately and rigorously? 

Reviewer #2: Yes

Reviewer #3: Yes

4. Have the authors made all data underlying the findings in their manuscript fully available?

Reviewer #2: Yes

Reviewer #3: Yes

5. Is the manuscript presented in an intelligible fashion and written in standard English?

Reviewer #2: Yes

Reviewer #3: Yes

6. Review Comments to the Author

Reviewer #2: Dear authors, it is a nice effort by you to carry up this draft in such a manner.

my comments have been satisfactorily addressed.

Good luck to all of you.

Reviewer #3: The recommended changes have been incorporated in the manuscript.

7. PLOS authors have the option to publish the peer review history of their article (what does this mean?). If published, this will include your full peer review and any attached files.

Reviewer #2: **Yes: **Gaurav Vedprakash Mishra

Reviewer #3: **Yes: **Anshul Sood

---

## [Editor Report · Acceptance letter]

10 Dec 2024

PONE-D-24-28190R1 

PLOS ONE

Dear Dr. liu, 

I'm pleased to inform you that your manuscript has been deemed suitable for publication in PLOS ONE. Congratulations! Your manuscript is now being handed over to our production team.

Kind regards, 

on behalf of

Dr. Gurinder Kumar 

Academic Editor

PLOS ONE